# A Comparative Study of Machine Learning Models for Predicting Vessel Dwell Time Estimation at a Terminal in the Busan New Port

Jeong-Hyun Yoon [1] , Se-Won Kim [1] , Ji-Sung Jo [2] and Ju-Mi Park [1,*]

1   Department of Intelligent Mechatronics Engineering, Sejong University, Seoul 05006, Republic of Korea;
    dbswjdgus9@gmail.com (J.-H.Y.); sewonkim@sejong.ac.kr (S.-W.K.)
2   Port Research Division, Korea Maritime Institute, Busan 49111, Republic of Korea; jisungjo@kmi.re.kr
*   Correspondence: jumi.park@hanwha.com

**Abstract:** Container shipping plays a pivotal role in global trade, and understanding the duration that vessels spend in ports is crucial for efficient voyage planning by shipping companies. However, these companies often rely solely on one-way communication for required arrival times provided by terminals. This reliance on fixed schedules can lead to vessels arriving punctually, only to face berths that are still occupied, resulting in unnecessary waiting times. Regrettably, limited attention has been given to these issues from the perspective of shipping companies. This study addresses this gap by focusing on the estimation of dwell times for container vessels at a terminal in the Port of Busan using various machine learning techniques. The estimations were compared against the terminal's operational reference. To compile the dataset, a 41-month history of terminal berth schedules and vessel particulars data were utilized and preprocessed for effective training. Outliers were removed, and dimensions were reduced. Six regression machine learning algorithms, namely adaptive learning, gradient boosting, light gradient boosting, extreme gradient boosting, categorical boosting and random forest, were employed, and their parameters were fine-tuned for optimal performance on the validation dataset. The results indicated that all models exhibited superior performance compared to the terminal's operating reference model.

**Keywords:** vessel dwell time; machine learning; regression; berth plan; container vessel; container terminal

## 1. Introduction

### 1.1. Research Background

The emergence of steel shipping containers has positioned container shipping at the forefront of international trade, catalyzing globalization and amplifying trade flows. An impressive 80% of global trade in goods is facilitated through an extensive network of container vessels [1]. These vessels have steadily grown in size over time, with larger capacities contributing to enhanced operational efficiency [2]. However, the expansion of vessel dimensions necessitates extended port stays unless additional quay cranes are allocated to expedite berthing operations. This prolonged duration that vessels spend at ports, known as vessel port dwell time, has escalated in significance, particularly for shipping enterprises [3]. For shipping companies, vessel port dwell time dictates the vessel's departure and validates the projected time of arrival at its subsequent destination. This, in turn, guides strategic planning, risk mitigation, and efficient navigational preparations. On the terminal side, vessel port dwell time serves as a pivotal measure, reflecting the prowess of container terminals in rendering services and quantifying their operational efficiency.

Vessel port dwell time, also referred to as vessel turnaround time, vessel time in port or berth time, designates the period during which a vessel is stationed at a port. As illustrated in Figure 1, the nomenclature and process of vessel-related timestamps are employed at container terminals. The actual time of arrival (ATA) signifies the moment

a vessel approaches the port and comes to a stop for anchoring. Upon securing a berth, the vessel's arrival at the berth corresponds to the actual time of berthing (ATB). The ATB timestamp corresponds to the instant when the vessel completes mooring to the port's bitts. Subsequent to arrival, terminal personnel such as forepersons and stevedores board the vessel to validate its stowage plan and prepare for cargo handling operations. The actual time of work (ATW) denotes the juncture when the vessel commences lifting containers on and off the vessel, signifying the initiation of cargo handling operations. The actual time of (work) completion (ATC) signifies the point at which the vessel finishes loading or unloading designated containers, concurrently concluding terminal operations. When the vessel is poised for departure, piloted, and lines released, the timestamp corresponds to the actual time of departure (ATD). Intervals between these timestamps may entail idling periods, potentially attributed to factors such as terminal staff shift changes, plan revisions, and various contingencies. Vessel port dwell time is inherently defined as the duration between ATD and ATB, encapsulating the period during which the vessel remains at the port. Typically, vessel dwell time closely parallels the interval between ATD and ATB. Yet, deviations may arise if the quantities of loaded, unloaded, and shifted containers are modified during berthing. Furthermore, variations in time spans may manifest between ATB and ATW, as well as between ATC and ATD, contingent on circumstances and diverse factors, encompassing port labor union policies, pilot or tugboat association regulations, and localized weather conditions. Given the multifaceted nature of factors influencing vessel dwell time, conventional estimations grounded in cargo load-based assessments often prove inadequate.

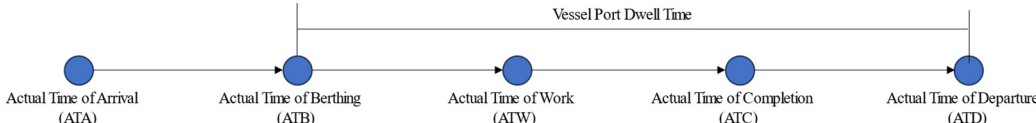

**Figure 1.** Terminology of vessel-related timestamps used at container terminals.

Consequently, the incorporation of machine learning techniques, capable of capturing nuances that elude human observation, is advocated. This underscores the rationale behind our hypothesis to harness machine learning models to outperform incumbent terminal-operated dwell time estimations. Machine learning algorithms hold the potential to uncover intricate patterns and relationships within extensive datasets that conventional statistical methods might overlook. This capacity becomes especially advantageous when confronted with intricate maritime systems and data.

*1.2. Scope of the Study and Research Area*

This study is predominantly centered on a container terminal situated within the Busan New Port precinct in South Korea. Renowned as the foremost port city in the nation and ranked 7th on a global scale [4], the port of Busan boasts a staggering container traffic volume of 22,078,000 TEUs (twenty-foot equivalent units) as of 2022. The port city hosts two principal clusters of container terminals, namely Busan Port and Busan New Port. Of these vibrant terminals, the Busan New Port Pier 1, also known as the Pusan New International Terminal (PNIT), emerged as the experimental site for this study. The PNIT encompasses three berths, each endowed with a 50,000-ton capacity. Leveraging authentic operational history data, our hypothesis was subjected to rigorous evaluation.

*1.3. Literature Review*

Previous studies have predominantly directed their attention toward terminal operations when delving into the domain of vessel dwell time. Some researchers have concentrated on modeling container cargo dwell time using simulation techniques. Studies [5–9] have tackled the estimation of container dwell time at terminal facilities, with a primary focus on enhancing cargo rehandling efficiency and identifying factors that impact overall terminal operations.

Concurrently, several works have investigated the factors influencing vessel turnaround time through analytical hierarchical process (AHP) analysis. Mapotsi [10] investigated factors responsible for extended vessel turnaround time at a dry bulk terminal. They formulated a multiple regression model encompassing diverse independent variables, including waiting time, idle periods, sailing and berth delays, alongside uncontrollable variables such as rain, wind, and visibility. Nyema [11] adopted a descriptive survey to gauge the significance of factors influencing vessel turnaround time by surveying approximately 500 employees at a container terminal. Rupasinghe, Sigera et al. [12] employed questionnaires across various stakeholder groups within terminal operations, utilizing AHP methodologies, and revealed determinants of delayed vessel turnaround time.

Studies have endeavored to characterize port productivity via vessel turnaround time analysis. Zhang and Kim [13] advocated the minimization of vessel turnaround time through optimized terminal quay crane (QC) allocation. Buhari, Ndikom et al. [14] developed a regression model predicting port revenue using cargo throughput and vessel turnaround time. Hong, Merk et al. [15] discerned factors influencing port performance in China's Shanghai port, employing statistical tools. Ming and Shah [16] examined operational processes within a petroleum terminal to mitigate vessel turnaround time. Jayaprakash and Gunasekaran [17] measured port performance using vessel turnaround time and service time, formulating a non-linear regression model that elucidates their interrelationships. Đelović and Mitrović [18] employed vessel turnaround time to evaluate berth productivity in a dry bulk cargo port.

Conversely, literature has also focused on identifying factors affecting vessel turnaround time across various terminals. Loke, Othman et al. [19] explored correlations between berth turnaround time and parameters such as total moves, discharge, and load quantities. Premathilaka [20] formulated a multiple linear regression model that integrates factors such as waiting time, container moves, quay crane deployment, crane intensity, and weather variables affecting vessel turnaround time. Dayananda Shetty, Gurudev et al. [21] conducted a detailed analysis of two-year vessel turnaround time data to identify driving factors. AV and Abijath [22] attributed increased vessel turnaround time at Cochin Port Trust to pre-detention and working time delays. Smith [23] analyzed port dwell times of container vessels having called at ports in the United States through automatic identification system (AIS) data, establishing a link between cargo volumes and vessel dwell time. Ducruet and Itoh [24] identified determinants impacting ship turnaround time through network analysis of international ports and nations.

The operational strategies adopted by shipping companies significantly influence fuel savings, given the substantial contribution of fuel costs to overall operating expenses. Research [25] indicates that fuel expenses constitute over 50% of total ship operating costs, necessitating meticulous fleet management. In this context, Moon and Woo [26] investigated the relationship between vessel dwell time in port and vessel operations at sea, considering operational costs and $CO_2$ emissions. Their work aligns closely with our study, emphasizing the significance of vessel dwell time in fostering efficient fleet management and operation. This concept can be encapsulated by the term "just-in-time arrival" (JITA) [27,28], also referred to as the "virtual arrival" (VA) [29] policy. The principle of the just-in-time arrival policy entails a streamlined operational protocol wherein a vessel, designated as A, adjusts its sea-going speed with consideration for the impending departure of another vessel, labeled B, from the berth where vessel A is destined to dock. This dynamic synchronization curtails excessive bunker consumption and associated emissions stemming from high-speed navigation. An illustrative instance of this approach can be found in the work of Yoon et al. [30], who leveraged historical vessel voyage trajectories to model forthcoming routes, augmenting the precision of vessel arrival time estimates at a container terminal with the overarching aim of implementing a robust JITA policy. It is pertinent to assert that endeavors of [26,30] are closely aligned with our objectives, centered on establishing streamlined interactions between vessels and terminals, thereby amplifying overall efficiencies and fostering data transparency through objective investigations.

Limited attention, however, has been accorded to estimating vessel dwell time from a carrier perspective, particularly using machine learning-based regression models. One notable study by Mokhtar and Shah [31] focused on predicting vessel turnaround time at a container terminal using a multiple linear regression model. Nevertheless, this study employed only a month's worth of data for model development, potentially limiting its representation of trends. Despite their insightful findings, follow-up studies in this area have remained scarce over the past 17 years. As highlighted by their work [31], vessel turnaround time is influenced by factors such as quay crane allocation and cargo handling, yet the adoption of more advanced methodologies is warranted.

Engaging personnel from a prominent container shipping company reveals the significance of having accurate vessel arrival and departure information. Anticipating port congestion using real-time data like the automatic identification system data could facilitate proactive management of port situations [32]. Presently, communication between ports and carriers is primarily one-way, with carriers adhering to the terminal's schedule. Establishing a two-way communication channel could enable vessels to adjust their speed at sea, thereby mitigating the need for excessive waiting time and contributing to fuel savings and emission reductions.

### 1.4. Research Objective

The primary objective of this study is to construct a comprehensive multiple regression model for the precise estimation of vessel port dwell time, employing the prowess of machine learning models. This model's effectiveness will be measured against the backdrop of prevailing terminal operations. In pursuit of this aim, the researchers embarked on the creation of six distinct machine learning models designed for regression analysis. Through an exhaustive process of grid searching, the optimal training configuration was identified, yielding the lowest error metrics across the dataset. Subsequently, the model exhibiting the best performance was systematically juxtaposed against the reference model's error metrics. This rigorous comparison sought to validate the superior efficacy of the proposed model and ascertain its potential as a robust tool for estimating vessel departure times, thus empowering container shipping enterprises with a reliable predictive framework.

### 1.5. Contributions

This study makes several significant contributions to the container carrier dwell time forecast:

1. The machine learning-based regression mode: a pioneering aspect of this study lies in the development of a novel machine learning-based regression model. This model was meticulously trained using comprehensive historical data on berth schedules, spanning an impressive 41-month duration. This extensive dataset served as the foundation for training and rigorously validating the model's predictive capabilities.
2. Enhanced voyage planning and terminal operations: the outcomes of this research offer tangible benefits not only to shipping companies but also to terminal operators. By enabling more effective voyage planning for shipping firms, the study facilitates streamlined interactions between vessels and terminals, reminiscent of the concept of the just-in-time arrival policy. This alignment fosters improved overall operational efficiency.
3. Efficiency with simple data: remarkably, this study achieved commendable results using a straightforward and basic dataset consisting of previous berth schedules and vessel particulars. The model's performance surpasses that of the reference model, underscoring the effectiveness of its approach.

## 2. Materials and Methods

### 2.1. Research Flow

The schematic diagram in Figure 2 illustrates the development of machine learning (ML) models designed to predict container vessels' dwell time at the port. The process

commenced with the collection of ten years' worth of berth schedule history data from a container terminal. In conjunction with this, pertinent particulars data for the vessels in berth were obtained through web crawling. Given the initial dataset's inherent imbalance, the authors opted to utilize only the most recent three years of data to enhance the model's performance. Subsequently, a new dataset was meticulously curated for training the ML models.

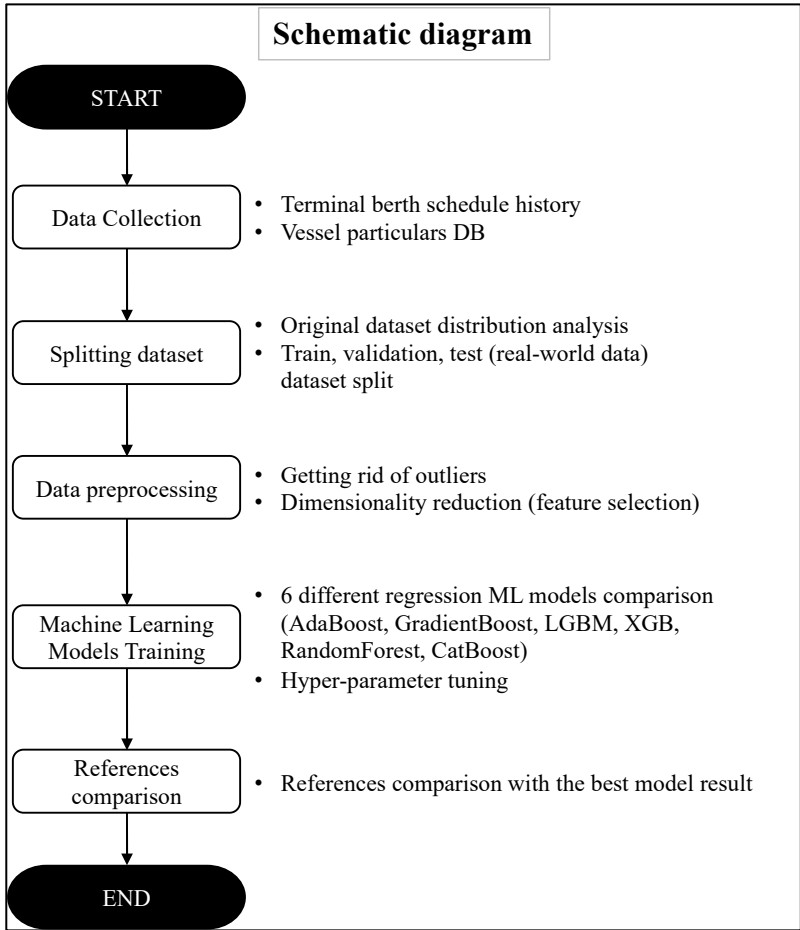

**Figure 2.** Research schematic diagram.

Before embarking on model training, several data preprocessing techniques were judiciously employed. These techniques encompass outlier removal, feature engineering, and dimensionality reduction. In total, six different ML models were considered for the regression task: adaptive boosting (AdaBoost), gradient boosting, light gradient boosting machine (LGBM), extreme gradient boosting (XGB), categorical boosting (CatBoost) and random forest.

To fine-tune the models' performance, a grid search methodology was applied to optimize their hyperparameters. Ultimately, the outcomes generated by the best-performing model were rigorously compared with reference data derived from terminal operations.

### 2.2. Dataset Configuration

#### 2.2.1. Data Collection

A multitude of container terminals readily provide their berth plans or schedules, often accessible through their websites. Fortunately, many terminals offer not only current berth schedules but also historical ones spanning various durations. A visit to one such website affiliated with a terminal in the Busan New Port, South Korea, facilitated the retrieval of years' worth of historical berthing plans (https://www.pnitl.com/infoservice/

vessel/vslScheduleList.jsp, accessed on 30th August 2023). In our quest for data richness, we gathered an extensive 12,153 rows of historical berth data from this source, equivalent to a decade of records. A representative sample of the berth schedule can be seen in Table 1.

**Table 1.** Berth schedule history sample.

| Berth | Company | Voyage | Vessel * | Time of Berth | Time of Departure | Loading Qty | Discharging Qty | Shifting Qty |
|-------|---------|--------|----------|---------------|-------------------|-------------|-----------------|--------------|
| T2(P) | BLA | $V_{246}005$ | $V_{246}$ | 1 January 2019 06:15:00 | 1 January 2019 16:00:00 | 139 | 211 | 0 |

* Specific vessel names and voyage details have been concealed for privacy reasons.

In Table 1, the "Berth" column denotes the assigned berth for the vessel, with the terminal housing a total of three berths: T1, T2, and T3. The notation "(P)" signifies that the vessel berthed on the 'port' side, which corresponds to the left side of the vessel, while "S" denotes 'starboard', signifying the right side. The "Company" field refers to the vessel's operating company name, represented by a three-letter code (defined in Appendix A). The "Voyage" code combines the vessel's unique code with the number of port calls in a given year. "Time of Berth" indicates the vessel's berthing time at the terminal, while "Time of Departure" signifies the moment of departure after cargo operations. The features "Loading Qty", "Discharging Qty", and "Shifting Qty" represent the quantity of containers handled between the vessel and the terminal, measured in TEUs (twenty-foot equivalent units). As elucidated in Figure 1, vessel dwell time is easily computed by subtracting "Time of Berth" from "Time of Departure".

Furthermore, a range of vessel particulars data, including length overall (LOA), width, gross tonnage, age, and capacity, were incorporated to enhance the ML model's performance and interpretability. These particulars were sourced through web crawling techniques and cross-verified with South Korean government public data repositories.

### 2.2.2. Data Exploration

During the exploratory data analysis phase, it became apparent that the original dataset exhibited a skewed and non-uniform distribution. Although initially appearing suitable for use, the dataset revealed a shift in the port's recent trends compared to historical records. To provide clarity, we visualized the total port calls and TEU counts at this terminal.

Figure 3 illustrates the total port calls at the terminal by year, while Figure 4 presents the total TEU counts over the same period. It is notable that annual port calls remained at their peak from 2014 to 2017, but a significant decline in port calls occurred from 2018 onwards. Conversely, Figure 4 indicates that TEU counts remained consistently high, even reaching their peak in 2018. This discrepancy can be attributed to the pandemic-induced port congestion, which led to a reduction in seaborne trade handling worldwide from 2020 to 2022. Notably, the trend of fewer port calls was driven by the increased visits of gigantic vessels (exceeding 10,000 TEU capacity).

According to JOC Group [33], the global ship turnaround time increased due to the influx of high-capacity vessels (over 10,000 TEU). This was driven by the assumption that larger vessels inherently enhance port productivity. As larger vessels handle more containers during a single port call, the number of vessels calling at the terminal decreases while container quantities remain substantial. This shift in port dynamics prompted us to focus primarily on the latest data, spanning over three years—41 months—for model training.

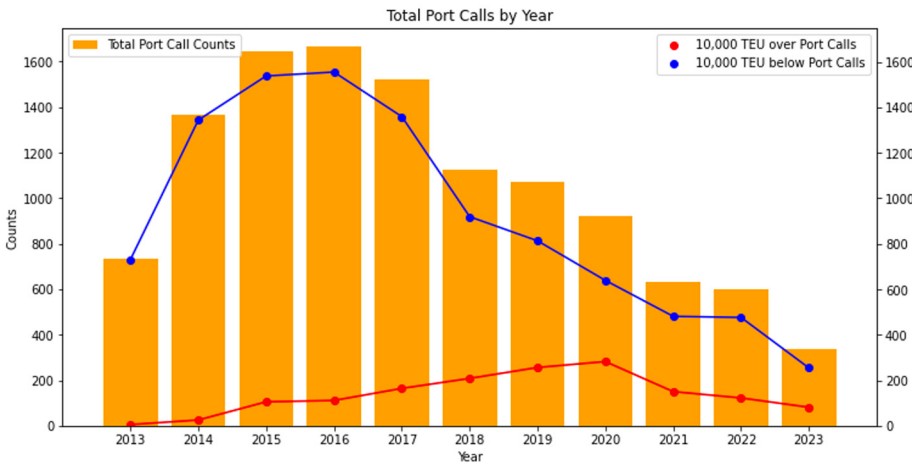

**Figure 3.** Total port calls by year.

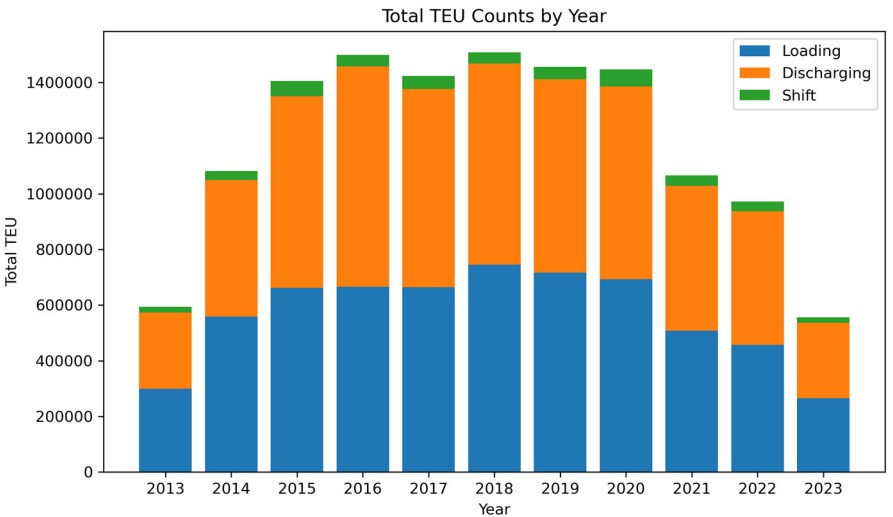

**Figure 4.** Total TEU (the number of containers moved) by year.

### 2.2.3. Splitting Dataset

The details of the dataset split are outlined in Table 2.

**Table 2.** Split dataset description.

| Dataset | Number of Rows | Proportion | Timespan |
|---|---|---|---|
| Train (randomly split) | 2653 | 67.68% | January 2019~August 2022 (32 months) |
| Validation (randomly split) | 664 | 16.96% | |
| Test | 597 | 15.25% | September 2022~June 2023 (9 months) |
| Total | 3914 | 100% | 41 months |

The dataset was composed of a total of 3914 rows of data, encompassing a 41-month duration. Within this dataset, the test data covered a period of 9 months, representing real-world data against which the model's predictions were assessed. The remaining 32 months of data were designated as the training set.

To avert the risk of overfitting, a common challenge in machine learning, where a model becomes excessively specialized to the training data, capturing not only underlying patterns but also noise, a validation dataset was thoughtfully created. This dataset was separated from the original training dataset and employed as a crucial tool to prevent overfitting. Both the new training and validation sets were randomly extracted from the initial training dataset.

### 2.3. Data Preprocessing

Data preprocessing is a pivotal phase in the preparation of data for machine learning models. It encompasses a range of techniques, including cleaning, handling missing values, transforming and selecting features, and addressing data imbalances. In this section, we meticulously executed a step-by-step preprocessing approach, encompassing the following stages: (1) removing outliers, (2) feature engineering, and (3) dimensionality reduction.

### 2.3.1. Removing Outliers

Outliers, those data points that markedly deviate from the general dataset pattern, can result from various sources, including data collection errors, measurement noise, or genuine extreme values. Addressing outliers is essential as they have the potential to distort both statistical analyses and machine learning model predictions. To mitigate the influence of outliers, we applied the interquartile range (IQR) method [34]. This method calculates the IQR as the difference between the 75th percentile (Q3) and the 25th percentile (Q1) of the data. The IQR effectively encapsulates the middle 50% of the data, reducing the impact of extreme values compared to the range defined by the minimum and maximum values. Outliers are typically identified through the following equations:

$$\text{Lower Outliers} < Q1 - (1.5 \times IQR) \tag{1}$$

$$\text{Upper Outliers} > Q3 + (1.5 \times IQR) \tag{2}$$

Figure 5 presents two box plots illustrating the difference in dwell time distribution in the dataset. The left figure displays raw data with outliers, while the right one exhibits the dataset after outlier removal using the IQR method. The central box in the plot represents the IQR. The top edge of the box represents the upper quartile, while the bottom edge represents the lower quartile. The length of the box corresponds to the spread of this middle 50% of the data. The orange horizontal line inside the box represents the median or the middle value of the dataset. The whiskers extend from the edges of the box to show the range of the data. They typically extend to a certain multiple of the IQR from the quartiles. Any data points beyond the whiskers are considered outliers like plotted as dots in the figure. Clearly, the raw data contained numerous outliers, while the processed data exhibited significantly less distortion in dwell time values.

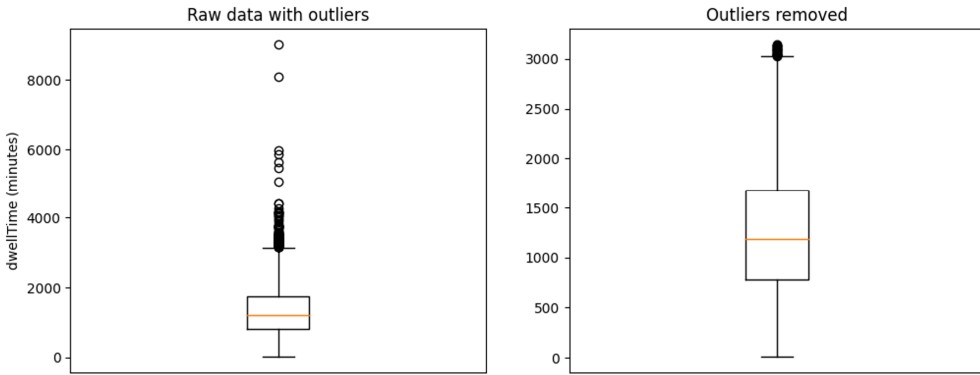

**Figure 5.** Raw data with outliers (**left**) and outliers removed (**right**).

Figure 6 offers another perspective on this comparison between raw data with outliers and processed data without outliers.

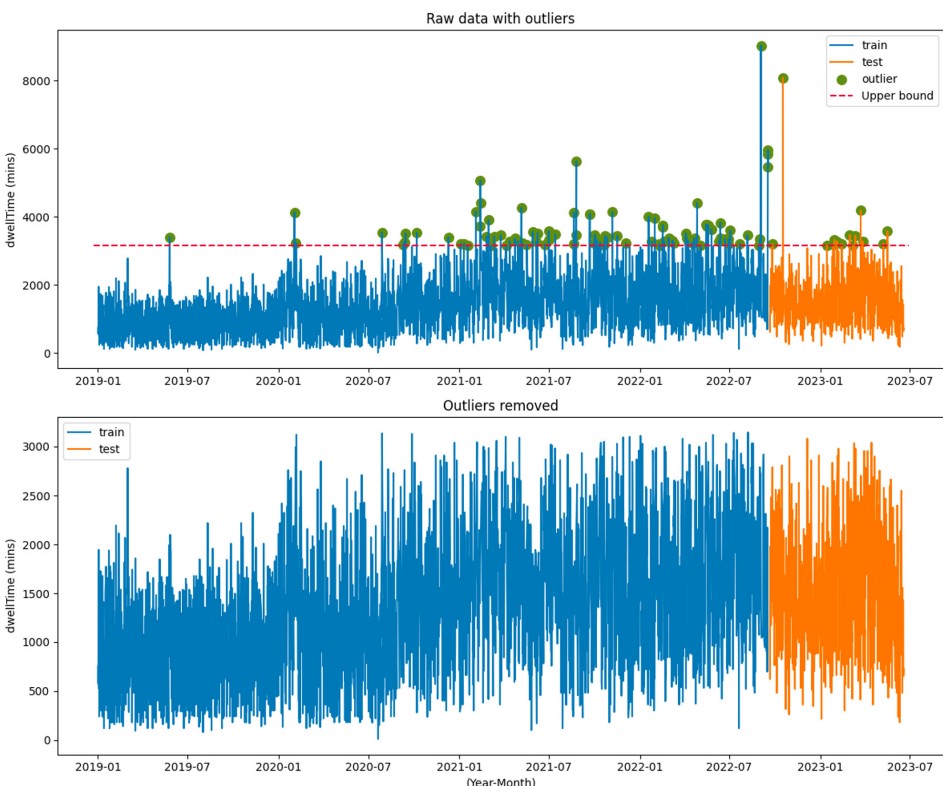

**Figure 6.** Raw data with outliers (**up**), Outliers removed (**down**).

### 2.3.2. Feature Engineering

Feature engineering entails selecting, creating, or transforming input variables to enhance a machine learning model's performance, often drawing on domain-specific knowledge. In this context, we identified and managed features through a systematic approach. For instance, we retained features related to company and berth dependencies, as we believed they played significant roles. Regarding the company feature, companies with fewer than 100 calls at this port during the dataset period were categorized as 'infrequent' to mitigate undue influence on the model. Additionally, to capture time series information such as seasonality and secular trends, we extracted year and month information from the 'Time of Berth' feature. Ship age was computed by subtracting the ship's year built from the year of berthing at the terminal, offering a representation of the age of the vessels. Lastly, the 'totalLoad' feature was derived by aggregating loading, discharging, and shifting quantities.

Figure 7 illustrates the distribution of categorical features, revealing a slight minority in the allocation of T3 berth. MSC (Mediterranean Shipping Company, Geneva, Switzerland) emerged as the most frequent company, followed by MAE (Maersk Sealand, Miramar, FL, USA). Figure 8 presents histograms depicting the distribution of numerical features. Features related to cargo quantity, such as loading, discharging, shifting, and total load, exhibit skewed trends, with fewer than 2000 cargos outweighing the larger ones. Notably, vessel dimension-related features, including gross tonnage (grossTon), length overall (LOA), width, and capacity, displayed random distributions, indicating a diverse range of vessels calling at this port, irrespective of their size and capacity. It is important to note that the dwell time feature exhibited a Gaussian distribution with some skew, suggesting that the dataset can be effectively trained without significant degradation of machine learning model performance.

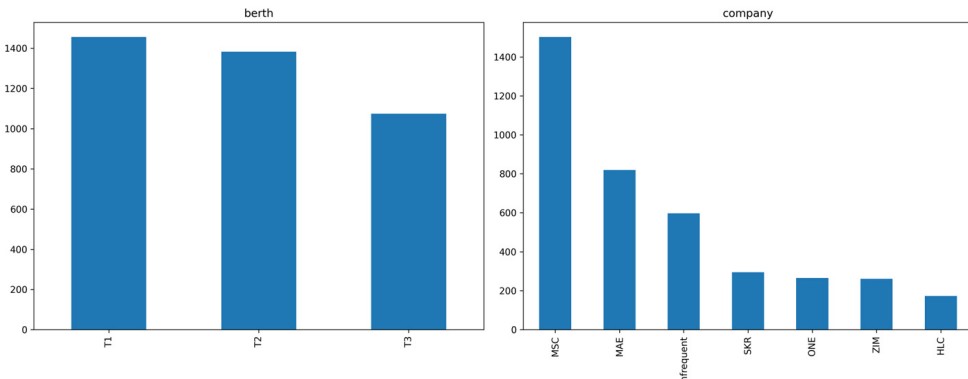

**Figure 7.** Categorical features distribution.

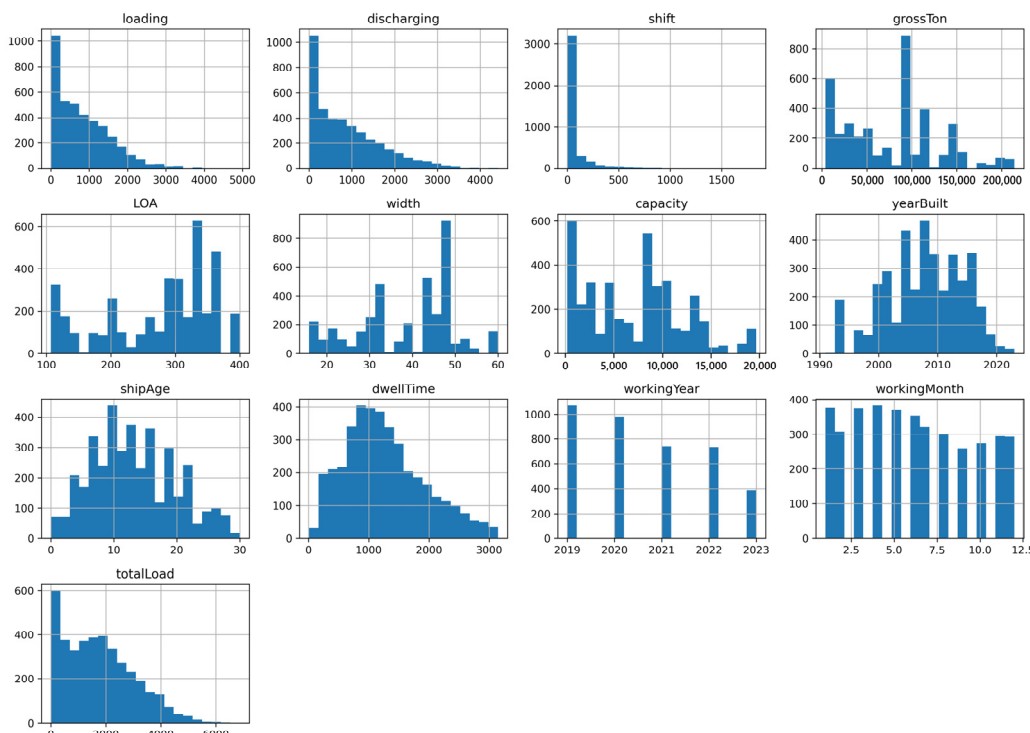

**Figure 8.** Numerical features distribution.

### 2.3.3. Covariate Shift Detection

Covariate shift in machine learning refers to a situation where the distribution of input covariates in the training data differs from the distribution of input features in the test or deployment data, which is likely to lead to the poor performance of the model prediction on the test data [35]. In this section, several covariate shift detection techniques are explained.

Figure 9 depicts a distribution plot comparing the training and test datasets. While each feature within both datasets did not exhibit significant imbalances, the plot highlights that the test dataset aligns well with the distribution of the training dataset. This observation provides evidence that the dataset is relatively unaffected by covariate shifts.

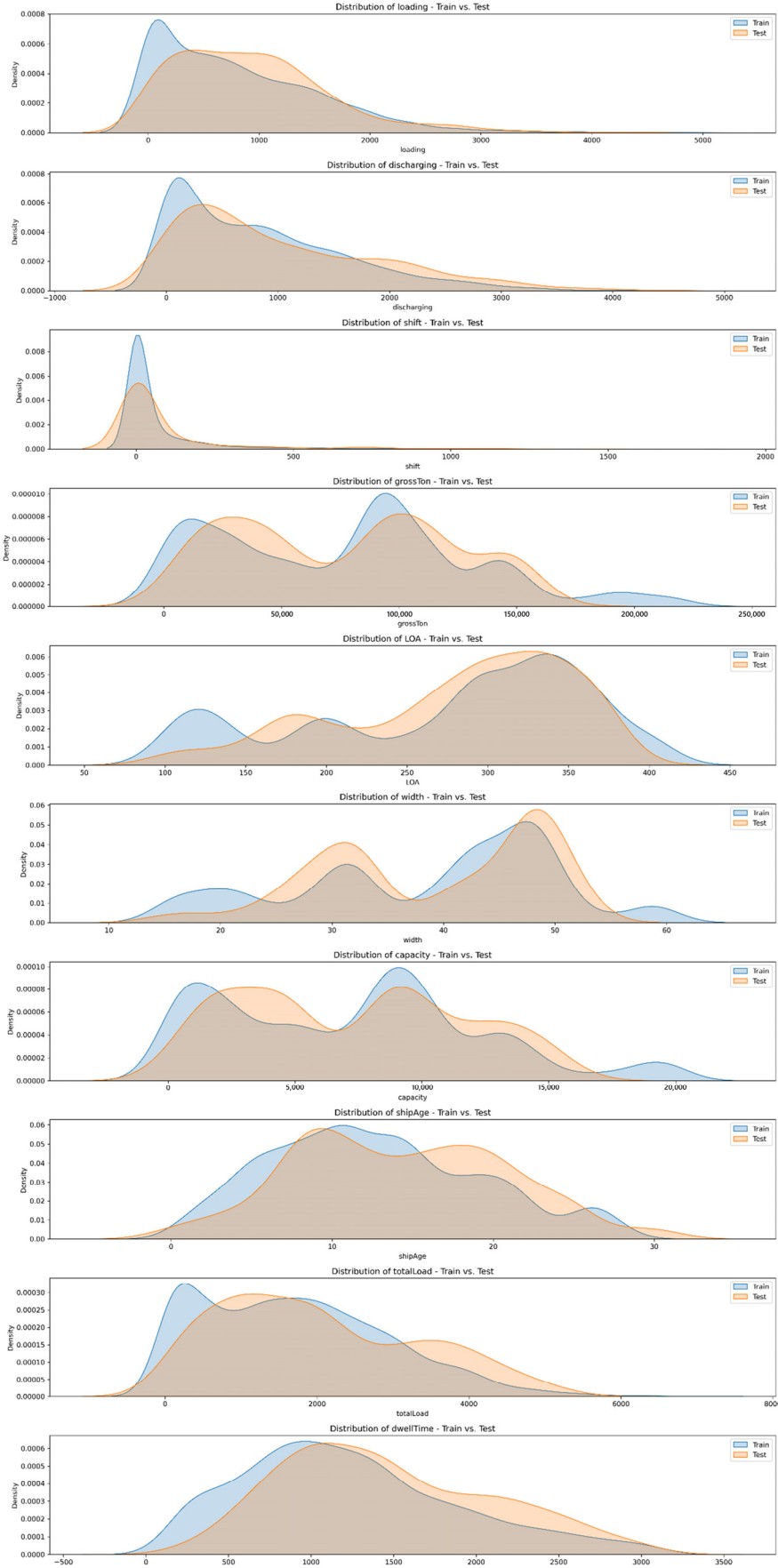

**Figure 9.** Distribution plot for train and test dataset.

Another technique employed to assess covariate shift is principal component analysis (PCA) [36]. PCA, a commonly used dimensionality reduction technique in machine learning and statistics, indirectly offers insights into potential covariate shifts by visualizing data in a reduced-dimensional space. PCA transforms the original data into a new coordinate system, with the first principal component capturing the most variance, the second component the second most, and so forth. This transformation enables us to focus on the most salient aspects of the data while reducing noise and redundancy.

Figure 10 illustrates the results of PCA visualization for covariate shift detection. After standardizing the data, we conducted PCA with two components. The objective of this analysis is to identify any significant divergence between the data points from the training and test datasets in this visualization, which could indicate the presence of a covariate shift. As shown in Figure 10, we observe the extent of overlap between the datasets, which serves as a key indicator of potential covariate shift. Notably, the visualization reveals that there is considerable overlap between the training and test datasets, signifying that the dataset is relatively immune to the effects of covariate shifts.

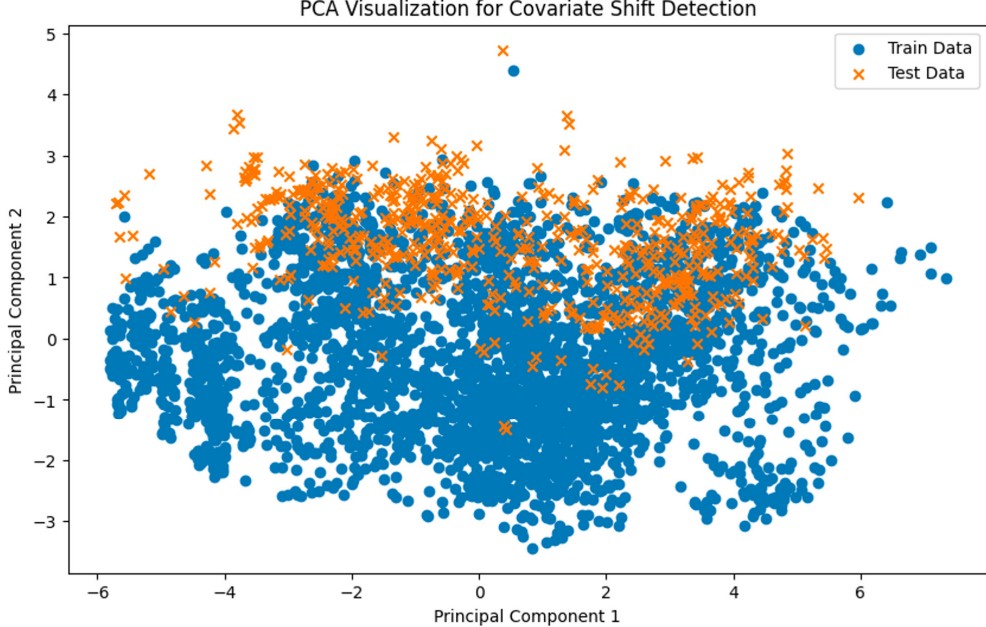

**Figure 10.** PCA visualization for covariate shift detection.

Moreover, the positions and spreads of the data points along the *x*-axis (Principal Component 1 or PC1) and *y*-axis (Principal Component 2 or PC2) in the plot can offer valuable insights into the covariance structure of the data. One notable observation is the horizontal alignment of the test dataset around values of PC2 ranging from 2 to 3, while the training dataset exhibits a wider horizontal distribution below PC2 of 3. This discrepancy in concentration along PC2 may indicate dissimilar covariance structures between the two datasets. The data's positioning implies that the test dataset has a higher density of data points along PC2 within the range of 2 to 3. In contrast, the training dataset demonstrates a more dispersed distribution along PC2 below the value of 3. This difference in concentration along PC2 suggests potential variations in covariance patterns between the two datasets. In practical terms, this suggests that the test dataset may exhibit stronger correlations or relationships captured by PC2 within the range of 2 to 3, which are less pronounced in the training dataset. This finding could imply the presence of certain patterns or relationships unique to the test dataset, as represented by PC2, which are not as prevalent in the training data.

The final measure to triple-check its covariate shift is a method called t-distributed stochastic neighbor embedding (t-SNE) [37]. The main idea behind t-SNE is to map each data point from its original high-dimensional space to a lower-dimensional space while

preserving the pairwise similarities between data points. It does this by modeling the probability distributions of pairwise similarities in both high- and lower-dimensional spaces. The algorithm minimizes the difference between these distributions, aiming to keep similar data points close to each other and dissimilar ones farther apart in the lower-dimensional representation. The method is also a dimensionality reduction technique commonly used for visualizing high-dimensional data in a lower-dimensional space. t-SNE is frequently used for visualizing high-dimensional data in a lower-dimensional space, making it an effective choice for detecting covariate shifts, especially given our dataset's 29 features.

Figure 11 showcases the t-SNE-driven plot for covariate shift detection, revealing the distribution of data points from the training and test datasets in the t-SNE space. Both datasets are clustered together, indicating similar characteristics. In summary, employing these measures for detecting covariate shifts provides strong evidence that the dataset exhibits consistent characteristics across its entirety.

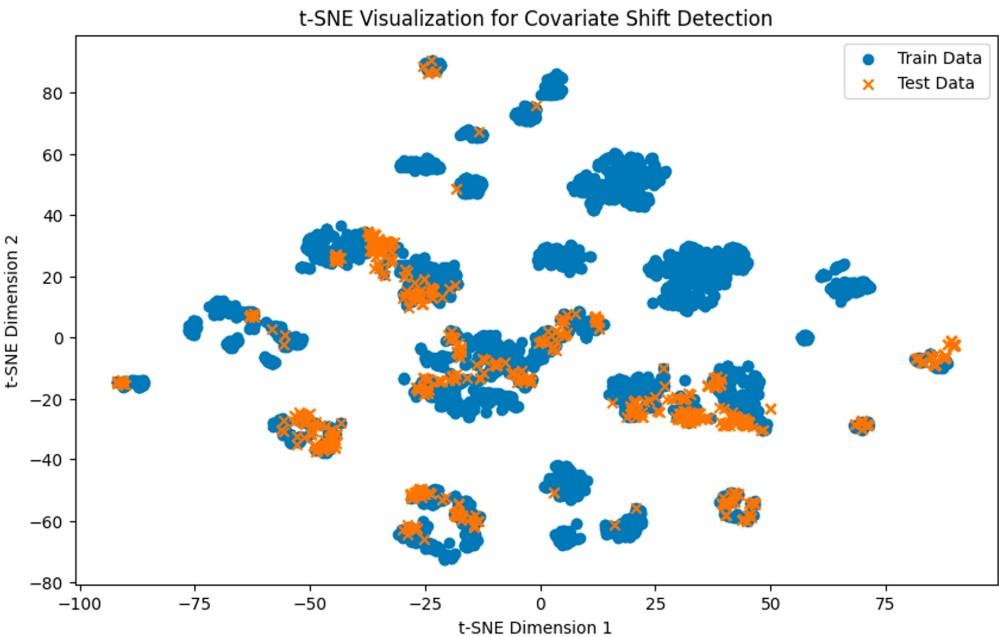

**Figure 11.** t-SNE visualization for covariate shift detection.

### 2.3.4. Dimensionality Reduction

Dimensionality reduction is a technique used in machine learning to reduce the number of features or variables in a dataset while retaining as much relevant information as possible. High-dimensional datasets with many features can often suffer from issues like the curse of dimensionality, increased computational complexity, and increased risk of overfitting. To prevent those issues, we demonstrated a feature selection technique that involves selecting a subset of the original features and informative features for the given problem. Some common feature selection methods include filter methods [38], wrapper methods [39], and embedded methods [40].

Each of these methods presents its own advantages and disadvantages. Filter methods employ statistical measures or heuristics to rank features based on their individual relevance to the target variable [38]. However, the mere selection of highly relevant features through filter methods does not guarantee their sustainability for the model. On the other hand, wrapper methods utilize a specific machine learning algorithm to assess subsets of features [39]. They involve training and evaluating models with various feature subsets to identify the optimal set of features. While effective, wrapper methods necessitate multiple model training iterations and are susceptible to overfitting, particularly when the dataset is small. In contrast, embedded methods combine the strengths of both filter and wrap-

per methods by integrating feature selection into the model training process [40]. These methods evaluate feature relevance within the context of the model's learning process, potentially leading to a more balanced feature selection that aligns with the model's generalization capacity. Moreover, embedded methods typically exhibit computational efficiency compared to wrapper methods, as they perform feature selection as an integral part of the model training process, reducing the need for multiple model training iterations.

One of the classes in the scikit-learn Python library, SelectFromModel (SFM), was employed to demonstrate embedded methods. Initially, we used a random forest regressor model as the reference model. After fitting the model with the training dataset, SFM was applied to extract crucial features. The determination of feature importance in tree-based models is grounded in the concept of a feature's contribution to reducing impurity in the nodes of trees. We employed the Gini Importance (GI) metric [41] to assess feature importance in the random forest model, as described in Equations (3) and (4).

$$G = \sum_{i=1}^{C} p(i) \cdot (1 - p(i)) \tag{3}$$

$$GI_i = \sum_{t \text{ in all trees}} \frac{n_t}{N} \cdot (G \text{ before split} - G \text{ after split}) \tag{4}$$

In Equation (3), $G$ is Gini Impurity, while $p(i)$ is the probability of picking a datapoint with class $i$ of the total classes $C$ [42]. In Equation (4), $n_t$ represents the number of samples that reach node $t$, while $N$ denotes the total number of samples in the dataset. $G \text{ before split}$ corresponds to the Gini impurity of the node before the split, while $G \text{ after split}$ signifies the weighted average of the Gini impurities of the two child nodes following the split. In simpler terms, the Gini Importance of a feature is the sum of reductions in Gini impurity observed when that feature is used for splits across all trees in the random forest. Features that consistently lead to purer nodes (i.e., lower Gini impurity) after a split are deemed more important. Additionally, since Gini importance values are usually normalized to add up to 1 or 100, we have normalized them to sum up to 1.

During training, the criterion employed was a mean squared error, with the threshold set to the median and a random state of 42. Following the fitting process, the R-squared ($R^2$) score for the model was 0.81 for the validation set and 0.72 for the test set.

Table 3 displays the feature importance results from the trained random forest model, where the threshold for feature importance was set at $2.116896 \times 10^{-3}$. In the original dataset, there were 29 features, but after filtering, only 15 remained.

**Table 3.** Feature importance.

| Feature | Importance |
|:---:|:---:|
| totalLoad | $8.035243 \times 10^{-1}$ |
| workingYear | $7.270265 \times 10^{-2}$ |
| discharging | $1.888127 \times 10^{-2}$ |
| loading | $1.803106 \times 10^{-2}$ |
| workingMonth | $1.556914 \times 10^{-2}$ |
| shift | $1.272530 \times 10^{-2}$ |
| LOA | $9.228835 \times 10^{-2}$ |
| capacity | $8.053313 \times 10^{-3}$ |
| grossTon | $7.731356 \times 10^{-3}$ |
| shipAge | $6.373691 \times 10^{-3}$ |
| width | $6.099715 \times 10^{-3}$ |
| yearBuilt | $5.785803 \times 10^{-3}$ |
| company_MAE | $2.269882 \times 10^{-3}$ |
| company_MSC | $2.165225 \times 10^{-3}$ |
| berth_T1 (Median) | $2.116896 \times 10^{-3}$ |

**Table 3.** *Cont.*

| Feature | Importance |
|---|---|
| berth_T2 | $1.906887 \times 10^{-3}$ |
| company_COH | $1.737024 \times 10^{-3}$ |
| berth_T3 | $1.385001 \times 10^{-3}$ |
| company_ZIM | $8.983881 \times 10^{-4}$ |
| company_ONE | $7.757470 \times 10^{-4}$ |
| company_SKR | $6.906458 \times 10^{-4}$ |
| company_HLC | $3.933310 \times 10^{-4}$ |
| company_HMM | $2.815237 \times 10^{-4}$ |
| company_COS | $2.066013 \times 10^{-4}$ |
| company_infrequent | $1.809573 \times 10^{-4}$ |
| company_HAS | $1.498511 \times 10^{-4}$ |
| company_OOL | $1.069325 \times 10^{-4}$ |
| company_BLA | $2.766817 \times 10^{-5}$ |
| company_DJS | $9.507312 \times 10^{-7}$ |

*2.4. Machine-Learning Models for Regression Tasks*

Machine Learning (ML) is about teaching algorithms to recognize patterns in data and use those patterns to predict outcomes or classify information [43]. In pursuit of even more accurate predictions, ensemble learning techniques like bagging and boosting have emerged as powerful strategies. Bagging, short for "bootstrap aggregating", assembles predictions from multiple instances of a single model, each trained on a different part of the data, effectively reducing overfitting and enhancing predictive stability [43]. On the other hand, boosting, a sequential learning process, constructs a sequence of models, each focusing on correcting the mistakes of its predecessor. This iterative approach ensures a refined and robust final model capable of capturing intricate relationships within the data [43]. In this subsection, an array of ensemble learning techniques, such as adaptive boosting, gradient boosting, light gradient boosting machine, extreme gradient boosting, categorical boosting and random forest, were briefly explained. All models presented in this study were used to compare their estimation performances.

2.4.1. AdaBoost Regressor [44]

Adaptive boosting, or AdaBoost, is an ML algorithm designed for both classification and regression tasks. AdaBoost is particularly effective at improving the performance of weak learners by sequentially combining their predictions to create a strong overall model.

After training a weak learner on the training data, it gives more weight to the misclassified data points, effectively focusing on the mistakes made by the previous learners. Moreover, the predictions of all weak learners are combined with weights based on their performance, while learners with higher accuracy or lower loss are given more weight. Misclassified points receive higher weights, making them more likely to be correctly classified in the next iteration. The final prediction in AdaBoost is made by combining the predictions of all weak learners, with each learner's contribution being weighted based on its accuracy.

The prominent feature of AdaBoost lies in its adaptability. It adjusts its focus on problematic data points as the iterations progress, allowing it to concentrate on the cases where previous models struggled. This adaptiveness leads to a strong ensemble model that can outperform individual weak learners.

2.4.2. GradientBoost [45]

Similar to AdaBoost, gradient boosting is an ensemble learning model used for both classification and regression tasks. However, unlike AdaBoost, gradient boosting builds a sequence of models where each new model tries to correct the mistakes made by the previous ones in a more systematic and data-driven manner. The term "gradient" refers

to the use of a gradient of the loss function when building each sequential model in the ensemble.

An initial model is trained on the data, and the next model is trained to predict the residuals, which are differences between actual and predicted values, of the previous model. It then combines the predictions of all models, and the predictions from the new model are added to the previous predictions, incrementally refining the overall prediction. The previous process is repeated, with each new model aiming to further reduce the errors made by the ensemble of previous models.

Gradient boosting is especially effective when dealing with complex relationships in data and high-dimensional datasets. It can capture nonlinear interactions and handle outliers better than some other algorithms.

### 2.4.3. LGBM Regressor [46]

LightGBM (light gradient boosting machine) is an efficient gradient boosting framework specifically designed for speed and performance. It is an open-source machine learning library developed by Microsoft. LightGBM is built to be faster and use less memory than other gradient boosting models while maintaining high accuracy or $R^2$ score.

Like other gradient boosting models, LightGBM builds an ensemble of weak learners to create a strong predictive model. It also uses a leaf-wise tree growth strategy, which prioritizes splitting nodes that lead to the largest reduction in loss. This strategy can speed up training but requires additional care to prevent overfitting. In addition, this machine uses histogram-based techniques to speed up the computation of gradients and Hessians during the tree-building process. This further contributes to its efficiency.

LightBGM is especially suitable for large datasets with many features, as its efficient algorithms can handle these situations more effectively than some other gradient boosting models.

### 2.4.4. XGB Regressor [47]

Like other gradient boosting techniques, XGBoost, short for Extreme Gradient Boosting, builds an ensemble of weak learners to create a strong predictive model. XGBoost enhances the concept of gradient boosting by incorporating regularization techniques and other optimizations for improved performance. As in others, after starting with an initial prediction, the residuals are calculated to improve these residuals with each new tree. Then, the model constructs a decision tree based on the features and the calculated residuals. The tree-building process involves selecting the best splits that minimize a loss function, which typically measures the difference between predicted and actual residuals. For regularization, it applies techniques such as L1 (Lasso) and L2 (Ridge) regularization to prevent overfitting. Once a new tree is constructed, its predictions are used to update the overall prediction. The predictions of all trees are combined, and each tree's contribution is controlled by the learning rate. The previous steps are repeated for a specified number of iterations, which means trees. Each new tree focuses on reducing the errors made by the ensemble of previous trees.

### 2.4.5. CatBoost Regressor [48]

The CatBoost regressor (categorical boosting regressor) is an ML algorithm designed for regression tasks built on the CatBoost framework. CatBoost is specifically optimized to handle categorical features effectively while also providing strong predictive performance for both categorical and numerical data. The CatBoost regressor is also one of the gradient boosting models and shares similarities with other gradient boosting techniques.

CatBoost uses an innovative approach called ordered boosting, which treats categorical features in a way that preserves their natural order and enhances the model's ability to capture their influence on predictions. It also employs a method to encode categorical features into numerical values during training, resulting in improved accuracy and efficiency. The regressor includes L2 regularization to control model complexity and prevent overfitting.

CatBoost has gained popularity in various machine learning competitions and real-world applications due to its ability to handle categorical features effectively and strong predictive performance. It is especially valuable when working with datasets that contain a mix of categorical and numerical features.

### 2.4.6. Random Forest Regressor [49]

The random forest regressor is an algorithm designed for regression tasks. It is a variant of the random forest algorithm, which is an ensemble learning technique. Random forest builds an ensemble of decision trees and combines their predictions to create a strong predictive model.

For each tree in the ensemble, a random subset of the data is sampled with replacement. This process is called bootstrap sampling, and it creates slightly different datasets for each tree. Each sampled dataset is then used to build an individual decision tree. These trees can be deep and complex and are trained to predict the target values based on the input features. In addition to sample data, random forest randomly selects a subset of features for each tree split. This randomness helps decorrelate the trees and reduce overfitting. Once all trees are constructed, their predictions are combined and averaged to obtain the final ensemble prediction.

The strength of the random forest regressor lies in its ability to handle complex relationships in the data, mitigate overfitting, and produce accurate predictions. Random forest is less sensitive to noisy data and outliers compared to individual decision trees.

### 2.5. Machine Learning Models Training

In this section, the crucial process of training machine learning models for the regression task at hand is explained.

### 2.5.1. Error Metrics

Error metrics play a crucial role in quantifying the performance of regression models and evaluating their predictive accuracy. In this subsection, we explore a range of error metrics employed to assess the quality of our trained models.

- Mean Absolute Error (MAE) [50]

$$MAE = \frac{\sum_{i=1}^{n}|Y_i - \hat{Y}_i|}{n} \tag{5}$$

where $\hat{Y}_i$ is the prediction of data point $i$, $Y_i$ is the true value of data point $i$ and $n$ is the total number of data points

Mean absolute error, or MAE, measures the average absolute difference between predicted and actual values. It is calculated by taking the average of the absolute differences between predicted and actual values. MAE is easy to interpret and gives equal weight to all errors, but it might not be suitable when outliers are present.

- Mean Squared Error (MSE) [50]

$$MSE = \frac{1}{n}\sum_{i=1}^{n}\left(Y_i - \hat{Y}_i\right)^2 \tag{6}$$

MSE, which stands for mean squared error, measures the average of the squared differences between predicted and actual values. It penalized larger errors more heavily than similar ones due to squaring. It is widely used but can be sensitive to outliers as their squared values dominate the overall error.

- Rooted Mean Squared Error (RMSE) [50]

$$RMSE = \sqrt{\frac{1}{n}\sum_{i=1}^{n}\left(Y_i - \hat{Y}_i\right)^2} \tag{7}$$

Root mean squared error, or RMSE, is the square root of the MSE, and it is often used to measure the typical size of errors in predictions. RMSE is in the same units as the original data, making it more interpretable. Like MSE, it is sensitive to outliers.

- R-squared (R$^2$ score) [50]

$$R^2 = 1 - \frac{RSS}{TSS} \tag{8}$$

where RSS is the sum of squares of residuals, TSS is the total sum of squares.

R-squared is a statistical metric used to evaluate the goodness of fit of a regression model. In the context of machine learning, regression models aim to predict a continuous target variable based on one or more independent variables (features). R-squared quantifies how well the independent variables explain the variability in the target variable. Yet, it tends to increase as more independent variables are added to the model, even if those variables are not truly meaningful. Thus, it does not account for model complexity. Additionally, R-squared can be misleading when used with nonlinear relationships, as it might still yield high values even if the model does not fit well.

- Adjusted R-squared (adjusted R$^2$ score) [50]

$$Adjusted\ R^2 = 1 - \left( \frac{(1 - R^2) \times (n - 1)}{n - p - 1} \right) \tag{9}$$

where $R^2$ is the regular R-squared value, $n$ is the number of observations (data points), and $p$ is the number of independent variables in the model.

The adjusted R-squared is often used to counter the issue of adding unnecessary variables to improve R-squared. The adjusted R-squared takes into account the number of independent variables in the model, penalizing the addition of unnecessary variables. This makes it a more reliable metric for assessing model fit while considering model complexity.

The key difference between R-squared and adjusted R-squared lies in how they handle model complexity. R-squared tends to increase with the addition of more variables, making it susceptible to overfitting. Adjusted R-squared, on the other hand, introduces a penalty for adding variables, providing a balance between model fit and simplicity. Therefore, in this study, adjusted R-squared was designated as a key metric to determine which model performs best and how much it is superior to others.

### 2.5.2. Hyperparameter Tuning [51]

Hyperparameter tuning plays a pivotal role in optimizing the performance of machine learning models. We explored the process of selecting the most appropriate hyperparameters to enhance the predictive accuracy and generalizability of our models. Through meticulous fine-tuning of parameters such as learning rate, the number of estimators, and maximum depth, our goal was to strike the delicate balance between underfitting and overfitting, thereby maximizing the effectiveness of our models in addressing real-world problems.

All machine learning models underwent training via the grid search cross-validation method. This technique systematically explores the best combination of hyperparameters by assessing model performance across various hyperparameter value combinations, employing cross-validation. Table 4 provides an array of hyperparameter values used for tuning each model.

The grid search method constructs a grid comprising all possible hyperparameter value combinations, where each combination represents a unique model configuration. For each configuration within this grid, the model undergoes training and evaluation through cross-validation, which entails dividing the dataset into multiple folds (e.g., $k$ folds [52]). The model trains on $(k - 1)$ folds and evaluates the remaining fold. This process repeats $k$ times, each time using a different fold as the evaluation set. In our experiment, we set the parameter $k$ to 10. After evaluating all configurations via cross-validation, each model identifies the configuration that yields the best performance metric, often determined by

the lowest error. Subsequently, the best hyperparameter configuration is selected before the final model is trained using the entire training dataset and the chosen hyperparameters. The results of hyperparameter tuning are presented in Section 3.2.

**Table 4.** Model parameters.

| Model | Parameter | Possible Value List |
|---|---|---|
| AdaBoost | n_estimators<br>learning_rate | [1, 2, 3, 4, 5, 6, 7, 8, 9, 10, 20, 30, 40, 50, 100, 150]<br>[0.001, 0.01, 0.1,0.2, 0.3, 0.4, 0.5, 0.6, 0.7, 0.8, 0.9, 1] |
| GradientBoost | n_estimators<br>learning_rate<br>max_depth | [10, 20, 30, 40, 50, 100, 150]<br>[0.0001, 0.001, 0.01, 0.1, 1, 10, 100]<br>[None, 1, 2, 3, 4, 5, 6, 7, 8, 9, 10] |
| LGBMRegressor | n_estimators<br>learning_rate<br>max_depth<br>num_leaves | [50, 100, 150]<br>[0.0001, 0.001, 0.01, 0.1, 1, 10, 100]<br>[None, 1, 2, 3, 4, 5, 6, 7, 8, 9, 10]<br>[2, 4, 6, 8, 10, 12, 15, 30, 31] |
| XGBRegressor | n_estimators<br>learning_rate<br>max_depth<br>booster | [10, 20, 30, 40, 50, 100, 150, 200, 250, 300]<br>[0.0001, 0.001, 0.01, 0.1, 1]<br>[1, 2, 3, 4, 5, 6, 7, 8, 9, 10]<br>[gbtree, gblinear, dart] |
| RandomForest | n_estimators<br>max_depth | [50, 100, 150, 200, 250, 300]<br>[1, 2, 3, 4, 5, 6, 7, 8, 9, 10] |
| CatBoostRegressor | iterations<br>learning_rate<br>max_depth<br>l2_leaf_reg | [50, 100, 150, 200, 250, 300]<br>[0.0001, 0.001, 0.01]<br>[None, 1, 2, 3, 4, 5, 6, 7, 8, 9, 10]<br>[0.2, 2, 5, 10, 20] |

## 2.6. Reference Model

In this subsection, we delve into the computation of the reference model's performance. At the container terminal under examination, the process of predicting and estimating berthing and departure times for vessels expected to arrive at the terminal occurs days or weeks in advance. Terminal operators take into consideration a multitude of factors, including estimated arrival times, total cargo handling quantities, and other determinants. This process forms the basis for creating a berthing plan [53]. However, the exact algorithms governing the reference model's operations remain somewhat elusive to outsiders, including the authors ourselves. Nevertheless, we were able to observe the predicted results, as all records were stored within the terminal operating system (TOS). Therefore, rather than defining the precise modeling intricacies, the authors regarded the reference model of the container terminal as a holistic representation of the prediction outcomes derived from the terminal's operations. In this study, we defined the reference model's outcomes using historical berth schedule data.

$Ref\_MAE$, or reference mean absolute error (MAE), is computed by aggregating the MAEs of numerous vessels within the research time period. It can be expressed simply by calculating the mean value of the estimated dwell time values for vessel $n$ using $l$ estimations and subtracting it from the actual dwell time for vessel $n$.

$$Ref\_MAE = \frac{\sum_{n=1}^{m} |(\frac{\sum_{s=1}^{l} estimated\ dwell\ time_s}{l}n) - actual\ dwell\ time_n|}{m}$$ (10)

In Equation (10), $m$ represents the total number of monitored vessels, $n$ is the index of the individual vessels, $l$ denotes the total number of reference estimations of vessel $n$, and $s$ is the timestamp index of vessel $n$.

While computing $Ref\_MAE$, we identified a few data points exhibiting significantly higher errors and chose to exclude those outliers to ensure a fair comparison with our models. Figure 12 comprises box plots that illustrate the differences in errors before and

after the removal of outliers. The left figure displays $Ref\_MAE$ without outliers following the application of the IQR method.

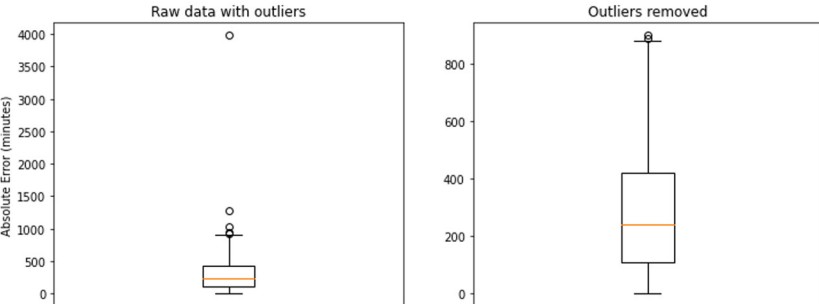

**Figure 12.** Reference results with outliers (**left**) and outliers removed (**right**).

Figure 13 presents a histogram depicting the errors generated by the reference model. Despite the removal of outliers, the reference model still exhibited relatively substantial errors, with an average error of 277 min.

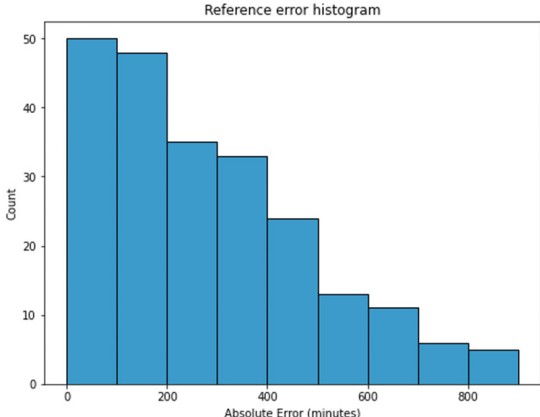

**Figure 13.** Reference error histogram.

## 3. Results

In this section, the results of this study are described in detail, covering model prediction results, hyperparameter tuning results and reference model and a comparison with the reference model.

### 3.1. Model Prediction Results

In this subsection, we compare the prediction results of six different regression machine learning models trained on the training dataset, considering both the validation and test datasets.

Figures 14 and 15 display the prediction results performed by the CatBoost model on the validation and test datasets, respectively. The *y*-axis in each figure represents the dwell time (in minutes) of vessels, which varies from under 500 min to over 3000 min. The *x*-axis represents the index of datasets. As previously mentioned in Table 2, the total rows in the validation set amount to 664; for the test set, there are 597 data points. The blue line represents the original values of the vessel's dwell time, which corresponds to the ground truth. The model's predicted dwell times for both datasets are plotted in orange. These plots demonstrate a strong overlap between the model's results and the original values, indicating a successful training process.

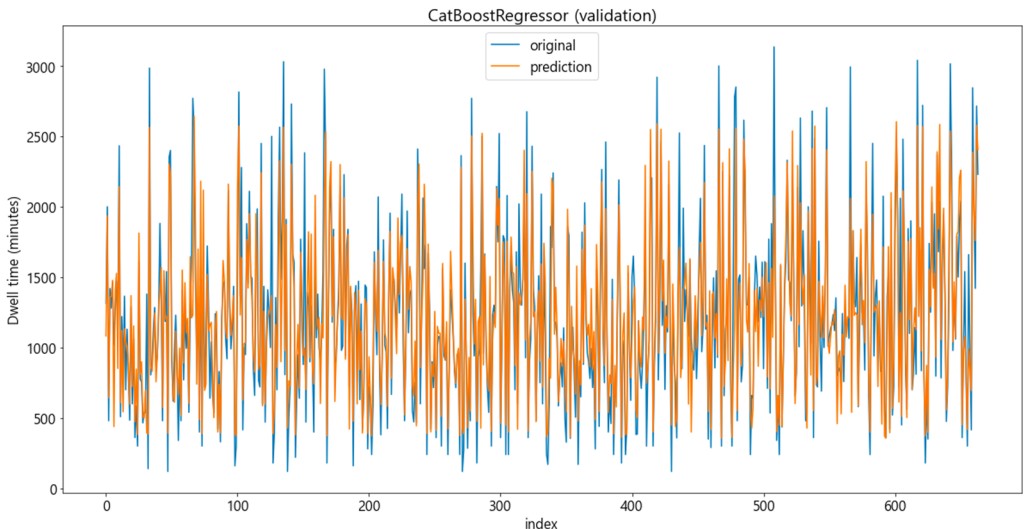

**Figure 14.** CatBoost model prediction result (for the validation dataset).

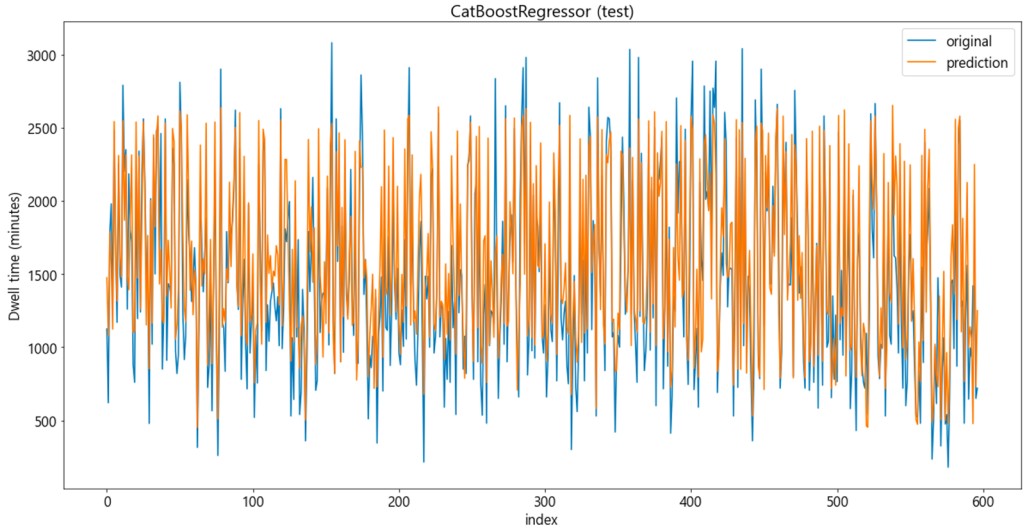

**Figure 15.** CatBoost model prediction result (for the test dataset).

Table 5 provides a comprehensive overview of the model prediction results for both the validation and test datasets. During the validation period, all models, except AdaBoost, performed similarly, yielding a mean absolute error (MAE) of around 180 min and an $R^2$ score of 0.82. Fortunately, the adjusted $R^2$ scores for these models were not significantly different from the ordinary $R^2$ scores, indicating that the feature selection and dimensionality reduction techniques effectively contributed to model performance. In general, model results on the validation dataset tend to be slightly superior to predictions on the test dataset, assuming that the models were trained without data leakage or data distribution mismatch. This slight difference is because test datasets are typically held separate from both training and validation, representing new, unseen data that the models have not encountered before. In this context, the model prediction results on the test dataset were slightly lower than those on the validation dataset, demonstrating no signs of data leakage or covariate shift. Models performed well on the test dataset, achieving adjusted $R^2$ scores of around 0.75 and a mean absolute error of approximately 250 min.

**Table 5.** Model prediction result.

| Dataset | Model | MSE * | RMSE * | MAE * | $R^2$ Score * | Adjusted $R^2$ * |
|---|---|---|---|---|---|---|
| Validation | AdaBoost | 94,785.37 | 307.87 | 236.79 | 0.77 | 0.76 |
| | GradientBoost | 69,828.51 | 264.25 | 182.20 | 0.83 | 0.82 |
| | LGBMRegressor | 74,334.73 | 265.02 | 182.47 | 0.83 | 0.82 |
| | XGBRegressor | 69,638.51 | 263.89 | 180.76 | 0.83 | 0.82 |
| | RandomForest | 73,022.97 | 270.23 | 186.28 | 0.82 | 0.82 |
| | CatBoostRegressor | 71,168.17 | 266.77 | 189.83 | 0.82 | 0.82 |
| Test | AdaBoost | 102,045.05 | 319.44 | 248.95 | 0.75 | 0.74 |
| | GradientBoost | 102,545.59 | 320.23 | 256.62 | 0.75 | 0.74 |
| | LGBMRegressor | 104,920.72 | 323.91 | 260.28 | 0.74 | 0.74 |
| | XGBRegressor | 101,342.16 | 318.34 | 255.24 | 0.75 | 0.75 |
| | RandomForest | 106,842.78 | 326.87 | 259.06 | 0.74 | 0.73 |
| | CatBoostRegressor | 94,295.66 | 307.08 | 248.47 | 0.77 | 0.76 |

* Values were rounded off to the nearest hundredth.

### 3.2. Hyperparameter Tuning Results

This subsection provides an overview of the hyperparameter tuning results for the six machine learning models used in this study.

Hyperparameters are parameters set before the training process that significantly influence the performance and behavior of a machine learning algorithm. As previously mentioned in Section 2.5.2, we utilized the GridSearchCV class in the scikit-learn library 0.20.4 for hyperparameter tuning. The results in Table 6 display the tuned hyperparameters for each machine learning model. The duration of the tuning process varied depending on the number of parameters involved in the grid search. These hyperparameters tuning experiments were conducted on a PC equipped with an Intel Core i7-10700F processor, 32 GB of RAM, and a solid-state drive. The programming tasks were performed using Python 3.7 on a Windows 11 machine.

**Table 6.** Hyperparameter tuning result.

| Model | Parameter | Hyper-Value | $R^2$ Score * | Tuning Duration |
|---|---|---|---|---|
| AdaBoost | n_estimators<br>learning_rate | 8<br>0.4 | 0.82 | 0:00:26 |
| GradientBoost | n_estimators<br>learning_rate<br>max_depth | 50<br>0.1<br>4 | 0.86 | 0:05:22 |
| LGBMRegressor | n_estimators<br>learning_rate<br>max_depth<br>num_leaves | 50<br>0.1<br>8<br>15 | 0.86 | 0:09:32 |
| XGBRegressor | n_estimators<br>learning_rate<br>max_depth<br>booster | 50<br>0.1<br>4<br>gbtree | 0.86 | 0:19:44 |
| RandomForest | n_estimators<br>max_depth | 250<br>7 | 0.85 | 0:01:03 |
| CatBoostRegressor | iterations<br>learning_rate<br>max_depth<br>l2_leaf_reg | 300<br>0.01<br>9<br>0.2 | 0.85 | 0:52:28 |

* $R^2$ scores were rounded off to the nearest hundredth.

### 3.3. References Result and Comparison

The exact algorithms of the reference model remain a black box, as it is challenging for outsiders, including the authors, to comprehend the intricacies of its operation. Nevertheless, the authors had access to the predicted results, as all records made by terminal

operators were stored in the terminal operating system (TOS). Therefore, instead of defining the precise modeling approach, the authors regarded the reference model of the container terminal as the aggregate prediction results derived from the terminal operation.

To clarify the results, consider this scenario: three days before vessel A's planned arrival, its berthing was originally scheduled for 10:00 on September 10th, with a departure time at 23:00 on the same date and an estimated dwell time of 11 h. However, for some reason, 24 h before its planned berthing, the terminal suddenly changed its berthing time to 13:00 on September 10th and its departure time to 04:00 on the next date. The estimated dwell time was also adjusted from 11 h to 15 h. When the vessel actually arrived and departure, the actual arrival time was at 12:00 on September 10th, and the departure occurred at 00:00 on September 11th, with an actual dwell time of 12 h. In this case, the terminal predicted its dwell time twice. The initial prediction absolute error was 1 h, and the subsequent error was 3 h. Thus, the mean absolute error for this case amounted to 2 h. The final reference model results were obtained by performing similar calculations for every vessel's case in the test dataset period from September 2022 to June 2023.

As calculated in Section 2.6, "Reference model", the reference model produced absolute errors with an average of 277.89 min for the test dataset period. As shown in Table 5, all models that made predictions on the test dataset outperformed the reference model.

## 4. Discussion

### 4.1. Result Analysis

In this study, we employed six different machine learning models, including adaptive boosting, gradient boosting, light gradient boosting, extreme gradient boosting, categorical boosting, and random forest, to predict vessel dwell times using historical berthing schedules. The results of these models are summarized in Table 5. Notably, all the model results on the validation dataset outperformed those on the test dataset.

This discrepancy in performance between the validation and test datasets, despite their similar sizes, may suggest differences in their underlying characteristics. The validation set was randomly sampled from the original training dataset spanning from January 2019 to August 2022, potentially inheriting some time-related trends and information from the training data. In contrast, the test dataset was selected from September 2022 to June 2023, making it distinct from the training data. This divergence implies that our models successfully learned from the training data without overfitting, as they generalized well to unforeseen data in the future.

Several factors contributed to the success of this study:

1.  Statistics and visualization approach: initially, the data collection process aimed for a larger dataset, assuming it would yield better results. However, during the analysis, we recognized the significance of considering trends in container movement and vessel capacity. For example, larger vessels with capacities exceeding 10,000 TEUs increased port calls at the terminal more than smaller vessels. Smaller vessels tended to have shorter stays and carried fewer containers. As terminal efficiency and container handling demands increased [33], more frequent port calls became prevalent. We also employed visualization tools such as Matplotlib and Seaborn, as illustrated in Figures 3, 4 and 7–9, to explore dataset distributions and trends. This proactive approach allowed us to filter and assess data based on distribution and trend characteristics.

2.  Data normalization: scaling data is a crucial step in machine learning to ensure consistent and effective model training. We utilized the standard scaler [54], a data scaling technique, to normalize the input features before training our models. Standard scaling transforms each feature into a mean of zero and a standard deviation of one. This technique benefits algorithms sensitive to feature scaling differences, promoting robustness, faster convergence, and better feature importance selection.

3.  Feature selection: feature selection played a significant role in refining our models. Features were removed based on their median threshold feature importance value,

calculated using the SelectFromModel (SFM) class in the scikit-learn library. This process helped us identify which features were most useful for model training. To assess the impact of this technique, we conducted a training experiment using only the top four features by importance, including "totalLoad", "workingYear", "discharging", and "loading". The results, as shown in Table 7, revealed that while the validation set results were similar to the initial model, those on the test set significantly differed. This observation suggests that factors influencing vessel dwell time, traditionally defined by various studies [10–12,19–22,24,31], may not be universally applicable. Instead, these factors may vary depending on terminal-specific policies and operational dynamics, highlighting the importance of selecting features that align with the specific terminal's historical data when estimating vessel dwell times.

4.  Hyperparameter tuning: to optimize model performance, hyperparameter tuning was conducted using the grid search cross-validation (GridSearchCV) technique. This method systematically explores a predefined hyperparameter grid to identify the optimal combination for each model. The adoption of hyperparameter tuning enhanced model performance and facilitated the identification of the best parameters for each model.

**Table 7.** Model results from the reduced features.

| Dataset (with Four Features) | Model | MSE * | RMSE * | MAE * | $R^2$ Score * | Adjusted $R^2$ * |
|---|---|---|---|---|---|---|
| Validation | AdaBoost | 92,436.03 | 304.03 | 228.14 | 0.77 | 0.77 |
| | GradientBoost | 76,503.75 | 276.59 | 193.59 | 0.81 | 0.81 |
| | LGBMRegressor | 74,334.73 | 272.64 | 189.74 | 0.82 | 0.82 |
| | XGBRegressor | 74,460.39 | 272.87 | 188.41 | 0.82 | 0.82 |
| | RandomForest | 76,809.47 | 277.15 | 192.07 | 0.81 | 0.81 |
| | CatBoostRegressor | 75,157.26 | 274.15 | 194.83 | 0.81 | 0.81 |
| Test | AdaBoost | 148,320.86 | 385.12 | 285.97 | 0.64 | 0.63 |
| | GradientBoost | 162,420.81 | 403.01 | 302.61 | 0.60 | 0.60 |
| | LGBMRegressor | 159,764.90 | 399.71 | 299.39 | 0.61 | 0.61 |
| | XGBRegressor | 163,696.40 | 404.59 | 304.05 | 0.60 | 0.60 |
| | RandomForest | 166,919.44 | 408.56 | 307.55 | 0.59 | 0.59 |
| | CatBoostRegressor | 157,256.97 | 396.56 | 293.88 | 0.61 | 0.61 |

* Values were rounded off to the nearest hundredth.

These factors collectively contributed to the success of our models in predicting vessel dwell times accurately and robustly. Our study demonstrates the importance of comprehensive data analysis, data scaling, feature selection, and hyperparameter tuning in enhancing the performance of machine learning models for complex tasks.

## 4.2. Additional Validation by Varying Test Periods

In this subsection, we present additional validation experiment results. While our models have demonstrated their effectiveness, we sought further verification through an additional experiment. As mentioned earlier, we trained the models on the training dataset and evaluated them on both the validation and test datasets. However, given that our dataset includes the 'Time of Berth' timestamp, we decided to divide the test dataset into several sub-groups based on specific time intervals. These intervals consisted of 8 weeks (2 months), 4 weeks (1 month), 3 weeks, 2 weeks, 1 week, 3 days, 2 days, and 1 day prior to the vessel's arrival. This segmentation aligns with the typical planning horizon of international carriers who schedule fleet voyages weeks in advance and make adjustments in the days leading up to arrival at the port.

The process of dividing the dataset proceeded as follows: taking the timestamp of the first row in the test data, which was 12 September 2022, we used it as a reference to define the upper boundary of the next sub-group based on the selected time interval. For instance, in the 8-week time interval, the upper boundary would be 12 October 2022. Any data

falling within this boundary constituted the first chunk of the test dataset. We repeated this process for all eight defined periods.

Figure 16 illustrates the model error results on various test periods, and Table 8 provides detailed information on these results, including mean absolute error (MAE) values for each sub-group. To arrive at these results, we calculated the MAE for each sub-group and then computed the mean values. Table 5, which showcases the test dataset results, revealed that the categorical boosting regressor (CatBoostRegressor) performed the best, with an MAE of 248.47 min, closely followed by adaptive boosting (AdaBoost) with an MAE of 248.95 min.

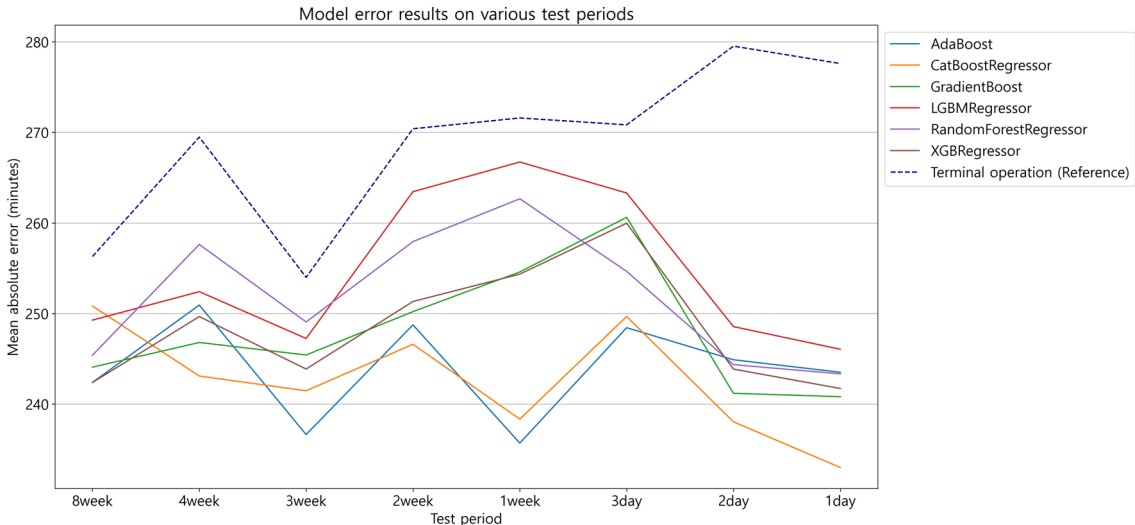

**Figure 16.** Model error results on various test periods.

**Table 8.** Model mean absolute error results on various test periods (unit: minutes).

| Model \ Periods | 8 Week * | 4 Week * | 3 Week * | 2 Week * | 1 Week * | 3 Day * | 2 Day * | 1 Day * |
|---|---|---|---|---|---|---|---|---|
| AdaBoost | 242.396 | 250.935 | 236.639 | 248.752 | 235.696 | 248.439 | 244.896 | 243.515 |
| GradientBoost | 244.078 | 246.804 | 245.425 | 250.212 | 254.591 | 260.625 | 241.191 | 240.822 |
| LGBMRegressor | 249.272 | 252.415 | 247.249 | 263.452 | 266.728 | 263.309 | 248.555 | 246.057 |
| XGBRegressor | 242.385 | 249.663 | 243.872 | 251.331 | 254.349 | 259.973 | 243.866 | 241.726 |
| RandomForest | 245.394 | 257.632 | 249.070 | 257.939 | 262.667 | 254.654 | 244.358 | 243.338 |
| CatBoostRegressor | 250.829 | 243.101 | 241.473 | 246.606 | 238.345 | 249.675 | 238.043 | 233.003 |
| Reference | 256.294 | 269.475 | 254.000 | 270.398 | 271.590 | 245.833 | 279.521 | 277.598 |

\* Values were rounded off to the nearest thousandth.

However, in this additional experiment, no significant differences were observed in the model results across various test periods. For instance, on the 8-week chunk, the extreme gradient boosting regressor (XGBRegressor) and AdaBoost exhibited the lowest errors, but on the 4-week chunk, both models experienced a sharp increase in errors. Conversely, the CatBoostRegressor outperformed the other models on four separate occasions: the 4-week, 2-week, 2-day, and 1-day periods. It is worth noting that there was a slight decreasing trend in performance over the different time intervals. Although the CatBoostRegressor is the most suitable model for this task, the selection of the best model may vary on different datasets due to the inherent randomness in machine learning and the specific data chosen.

### 4.3. Limitations of This Study

However, this study has some limitations. Firstly, our testbed focused on a single container terminal, which means that the model results are specific to this particular terminal. Additionally, we did not consider other types of vessels, such as general cargo, liquid natural gas (LNG), chemical tankers, or pure car and truck carriers (PCTCs). Furthermore, we could not incorporate real-time terminal-oriented data, such as quay crane allocation

plans, which could potentially enhance prediction accuracy. Moreover, the exact algorithms used by the terminal to estimate port dwell time remain undisclosed to us. Having access to these algorithms would have allowed for a more precise comparison between our models and the terminal's estimations.

Last but not least, there is still room for improvement in this ML modeling. Although all models trained in this study outperformed the reference model, a 30-min improvement, which represents about a 12% gain, may not be significantly superior to the reference model. Utilizing machine learning models can lead to more accurate estimates of vessel dwell time compared to terminal-provided estimates, reducing planning uncertainties and increasing operational efficiency. However, this improvement is contingent on employing better techniques and a variety of data sources to build a more sophisticated machine learning model.

## 5. Conclusions

### 5.1. Summary

This study aimed to estimate the dwell time of container vessels in the port using multiple machine learning models and techniques, comparing these estimations with those provided by the terminal's operational reference. We compiled a dataset using 41 months of terminal berth schedule history and vessel particulars data. Prior to training our models, we performed data preprocessing, including the removal of outliers and dimensionality reduction, to optimize the dataset for training. We trained six regression machine learning algorithms: adaptive boosting, gradient boosting, light gradient boosting, extreme gradient boosting, categorical boosting and random forest. The model parameters were fine-tuned to achieve the best results on the validation dataset. The outcome of this analysis revealed all of the machine learning models outperformed the reference model used at the terminal by using simple datasets and fewer resources.

### 5.2. Future Works

Future endeavors will address some of the limitations encountered in this study. We intend to expand our research to various terminals and vessel types, including pure car and truck carriers (PCTCs) and chemical carriers, as their port dwell times are known to be shorter compared to container vessels. Additionally, we will consider incorporating various types of data sources, such as automatic identification system (AIS) data and arrival documents (e.g., customs, immigration, and quarantine—CIQ), which may undergo changes upon a ship's arrival. These enhancements will contribute to a more comprehensive and accurate estimation of vessel dwell times.

**Author Contributions:** Conceptualization, J.-H.Y.; methodology, J.-H.Y.; software, J.-H.Y.; validation, S.-W.K., J.-S.J. and J.-M.P.; formal analysis, J.-H.Y., J.-S.J. and J.-M.P.; investigation, J.-H.Y.; resources, S.-W.K.; data curation, J.-H.Y.; writing—original draft preparation, J.-H.Y.; writing—review and editing, J.-H.Y., S.-W.K., J.-S.J. and J.-M.P.; visualization, J.-H.Y.; supervision, S.-W.K.; project administration, S.-W.K.; funding acquisition, S.-W.K. All authors have read and agreed to the published version of the manuscript.

**Funding:** This research has received partial support from the Korea Institute of Marine Science & Technology Promotion (KIMST), funded by the Ministry of Oceans and Fisheries (RS-2023-00256127) (50%). Additionally, it has been funded by Hanwha Ocean CO., LTD, as part of the Development of Vessel Just-in-time Arrival Algorithms Supporting Economic Sailing through Interactions between Smart Ships and Smart Ports project (project serial number: 20230608).

**Institutional Review Board Statement:** Not applicable.

**Informed Consent Statement:** Not applicable.

**Data Availability Statement:** Data sharing is not applicable to this article due to privacy and security concerns.

**Conflicts of Interest:** The authors declare no conflict of interest.

## Appendix A. Shipping Company Code

Table A1 shows a description of the company codes utilized at this terminal. The majority of these shipping companies are involved in the international shipment of container cargo. It is important to note that these company codes were designated by terminal personnel.

**Table A1.** Company code description.

| Company Code | Full Name |
|---|---|
| MSC | MEDITERRANEAN SHIPPING COMPANY S. A. (MSC) |
| MAE | MAERSK SEALAND |
| SKR | SINOKOR MERCHANT MARINE CO., LTD |
| ONE | Ocean Network Express (ONE) |
| ZIM | ZIM INTEGRATED SHIPPING SERVICES LTD |
| HLC | HAPAG-LLOYD AG |
| COS | CHINA OCEAN SHIPPING (GROUP) CO. |
| COH | COSCO SHIPPING KOREA CO. |
| HMM | HMM CO., LTD |
| HAS | HEUNG-A SHIPPING CO., LTD |
| OOL | ORIENT OVERSEAS CONTAINER LINE (OOL) |
| BLA | BEN LINE AGENCY |
| APL | AMERICAN PRESIDENT LINES., LTD. |
| DJS | DONGJIN SHIPPING CO., LTD |

## Appendix B. Correlation Heatmap

Figure A1 displays a heatmap illustrating the correlations between independent features present in the original dataset.

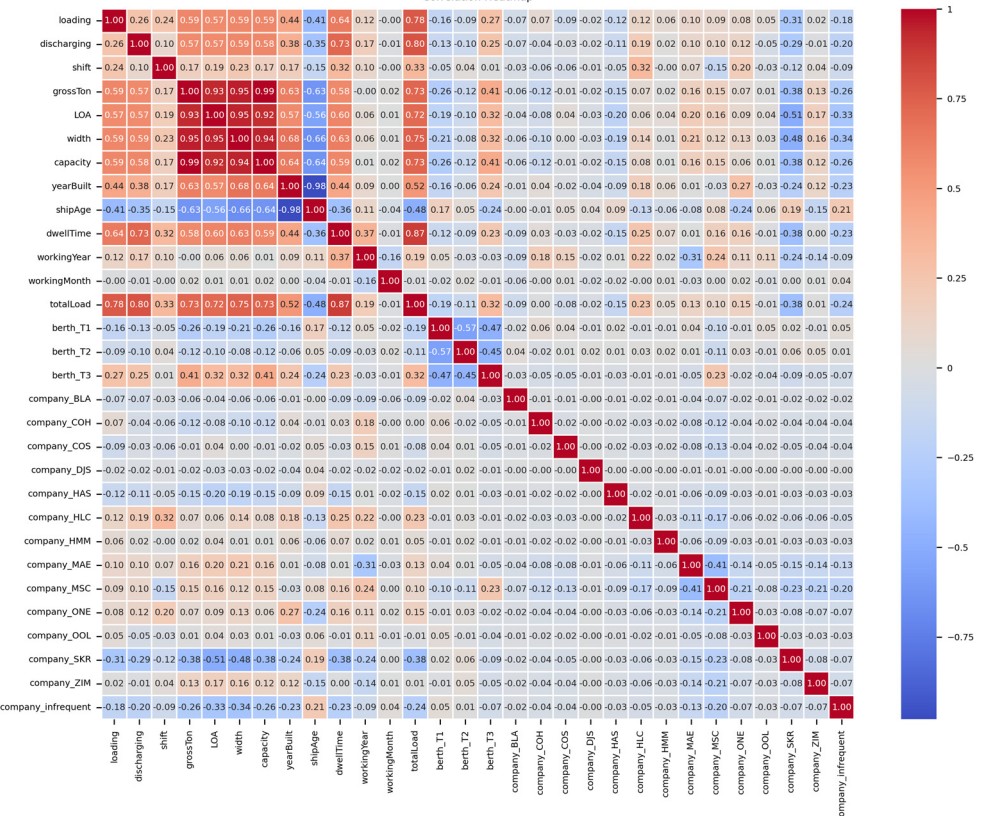

**Figure A1.** Correlation heatmap of independent features.

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
