# Peer review of "A Comparative Study of Machine Learning Models for Predicting Vessel Dwell Time Estimation at a Terminal in the Busan New Port"

_jmse, doi:10.3390/jmse11101846_

Round 1
Reviewer 1 Report
Authors have written a nice comparison of ML methods on some terminal dwell time data from a Busan terminal. The results seem to show a slight superiority for one ML model, and some benefit of the ML model over the unknown method used by the terminal operator.
The paper explains the methodology and the routines chosen for comparison quite clearly. Overall the paper is well-written.
I made a few notes in the attached pdf of the paper.
I am concerned that the routines were not that different overall, and even not that different from the unknown forecaster used by the terminal management. There are theoretical reasons for believing CatBoost to be better, because the data contains categorical variables. But how do we know that the MAEs obtained would not vary with different test and validation datasets on which the algorithms were trained?
This question could be addressed with more experimentation, and I am not proposing that authors do it here. Perhaps some caution could be inserted into the conclusions regarding this. ML algorithms are difficult to judge because of the randomization they use, and because the validation techniques might be sensitive to the specific data chosen.
The paper is a good example of how to write an ML paper. But caution in interpreting results must be shown.

English very good, just a couple of issues I found and marked.
Author Response
Thank you for your insightful and productive review. We sincerely appreciate your efforts.

Reviewer 2 Report
Dear Authors,
Thank you for the informative and actual paper you've submitted. I have a couple of things to share with you:
1. A big plus for the correlation heatmap 'illustrating the correlations between independent features present in the original dataset'.
2. Congrats for the results on the validation dataset that 'outperformed those on the test dataset'. I am sure you have strong arguments to support this fact.
3. I am not using any black box algorithms, and even those poorly documented I am used to avoid. Once again, I hope your sources are reliable enough!
4. Some common algorithms used for time-series forecasting are: Autoregressive-Integrated-Moving Average (ARIMA) and Long Short-Term Memory (LSTM). Why haven't you used them?
Thank you!
Author Response

(The authors gave the same response as above.)
